# Foot Revascularization Avoids Major Amputation in Persons with Diabetes and Ischaemic Foot Ulcers

**DOI:** 10.3390/jcm10173977

**Published:** 2021-09-02

**Authors:** Marco Meloni, Daniele Morosetti, Laura Giurato, Matteo Stefanini, Giorgio Loreni, Marco Doddi, Andrea Panunzi, Alfonso Bellia, Roberto Gandini, Enrico Brocco, José Luis Lazaro-Martinez, Davide Lauro, Luigi Uccioli

**Affiliations:** 1Department of Systems Medicine, University of Rome “Tor Vergata”, 00133 Rome, Italy; lauragiurato@yahoo.it (L.G.); andreapanunzi@gmail.com (A.P.); alfonso.bellia@ptvonline.it (A.B.); d.lauro@med.uniroma2.it (D.L.); luccioli@yahoo.com (L.U.); 2Department of Interventional Radiology, University of Rome “Tor Vergata”, 00133 Rome, Italy; danielemorosetti@hotmail.com (D.M.); roberto.gandini@fastweb.net (R.G.); 3Department of Radiology, Casilino Polyclinic, 00169 Rome, Italy; matteostefanini@hotmail.com; 4Department of Interventional Radiology, Sandro Pertini Hospital, 00157 Rome, Italy; giorgioloreni@hotmail.it (G.L.); marcododdi@gmail.com (M.D.); 5Diabetic Foot Centre, Abano Terme Polyclinic, 35031 Abano Terme, Italy; enbrocco@gmail.com; 6Instituto de Investigacion Sanitaria San Carlo Hospital, Complutense University of Madrid, 28040 Madrid, Spain; diabetes@ucm.es

**Keywords:** diabetes, diabetic foot ulcer, critical limb ischaemia, revascularization, wound healing

## Abstract

The study aims to evaluate the effectiveness of foot revascularization in persons with diabetic foot ulcers (DFUs) and below-the-ankle (BTA) arterial disease. Consecutive patients referred for a new active ischaemic DFU requiring lower limb revascularization were considered. Among those, only patients with a BTA arterial disease were included. Revascularization procedures were retrospectively analysed: in the case of successful foot revascularization (recanalization of pedal artery, or plantar arteries or both) or not, patients were respectively divided in two groups, successful foot perfusion (SFP) and failed foot perfusion (FFP). Healing, minor and major amputation at 12 months of follow-up were evaluated and compared. Eighty patients (80) were included. The mean age was 70.5 ± 10.9 years, 55 (68.7%) were male, 72 (90%) were affected by type 2 diabetes with a mean duration of 22.7 ± 11.3 years. Overall 45 (56.2%) patients healed, 47 (58.7%) had minor amputation and 13 (16.2%) major amputation. Outcomes for SFP and FFP were respectively: healing (89.3 vs. 9.1%, *p* < 0.0001), minor amputation (44.7 vs. 78.8%, *p* = 0.0001), major amputation (2.1 vs. 36.3%, *p* < 0.0001). Failed foot revascularization resulted an independent predictor of non-healing, minor amputation, and major amputation. Foot revascularization is mandatory to achieve healing and avoid major amputation in persons with ischaemic DFU and BTA arterial disease.

## 1. Introduction

Peripheral arterial disease (PAD) is a common complication of diabetes and can be documented approximately in 50% of persons with diabetic foot ulcers (DFUs) [1,2]. PAD in diabetic people is often bilateral, distal and shows a high rate of recurrence after peripheral revascularization [3,4,5]. The involvement of infra-popliteal vessels (anterior tibial artery, posterior tibial artery, peroneal artery) is a specific characteristic of PAD in diabetics [6,7]. It is well known as revascularization of infrapopliteal vessels is effective to avoid major amputation in patients with ischaemic DFUs [7,8], and recanalization for as many infrapopliteal arteries as possible increases the chance of limb salvage [9]. However, recently, a common involvement of below-the-ankle (BTA) arteries (plantar and pedal arteries) in patients with ischaemic DFUs has been documented [10,11].

In addition, BTA arterial disease denotes a more aggressive pattern of PAD and it appears to be an independent predictor of non-healing in patients with ischaemic DFUs [11,12]. In this specific case, it seems useful to address revascularization procedures to foot arteries and not only to infra-popliteal vessels. Nonetheless, to our knowledge there are few specific data on the effect of foot revascularization in patients with BTA arterial disease and ischaemic DFUs. Accordingly, this study aims to evaluate the impact of foot revascularization in people with ischemic DFUs and BTA arterial disease.

## 2. Materials and Methods

### 2.1. Patient Selection

This study is a retrospective study conducted at a single centre. Consecutive patients who were referred to a tertiary level Diabetic Foot Service (DFS) in Rome, Italy, since 2014 to 2019 for a new active ischaemic DFU requiring lower limb revascularization were considered in the study, while patients with neuropathic DFU, reduced life-expectancy (less than 6 months) and unsalvageable foot at admission were excluded (see Figure 1). Through a retrospective analysis of angiograms recorded during the revascularization procedures, only patients with a BTA arterial disease were included (see Figure 1). BTA arterial disease has been considered by the absence of foot blood perfusion (below-the-ankle) due to the stenosis and/or occlusion, single or multiple, of pedal artery and medial and lateral plantar arteries or plantar arch.

All patients included were managed by a pre-set limb salvage protocol following International Working Group on the Diabetic Foot (IWGDF) Guidance [13], including restoration of foot perfusion by lower limb revascularization, antibiotic therapy (and surgery if required) in the case of infection, offloading of affected limb, management of diabetes and comorbidities, ulcer debridement and local wound care based on the best evidence recommendations. Data were collected in a local database and retrospectively analyzed. Baseline demographic, clinical and ulcer features were recorded. The study has been approved and performed according to local ethics committee policy.

### 2.2. Clinical Assessment

Hypertension was defined in the case of current antihypertensive therapy; hypercholesterolemia was defined in the case of current statin therapy; ischaemic heart disease (IHD) was defined in the case of previous acute coronary syndrome or coronary revascularization, evidence of angina, significant changes on electrocardiography (above or under-levelling ST, q wave, inversion of T wave, new left bundle branch block). Heart failure (HF) was defined in the case of typical symptoms and signs of HF reduced left ventricular ejection fraction (LVEF) (<40%) or normal or only mildly reduced LVEF and elevated levels of brain natriuretic peptides (BNP > 35 pg/mL and/or NT-proBNP > 125 pg/mL) without dilated left ventricle (LV), associated with relevant structural heart disease (LV hypertrophy/left atrial enlargement) and/or diastolic dysfunction [14]. Cerebrovascular disease (CVD) was defined in the case of previous cerebrovascular ischaemia, previous carotid revascularization, or significant carotid artery disease (occlusion > 70%). Dialyzed patient was defined in the case of end-stage-renal-disease (ESRD) requiring renal replacement therapy.

### 2.3. Ulcer Characteristics

Ulcer characteristics reported at the time of presentation, and first assessment at the DFS. Deep ulcers were defined in the case of full thickness skin lesions, extending from the subcutaneous to tendon, muscle, or bone. Diagnosis of infection was defined according to IWGDF guidelines [13].

### 2.4. Vascular Assessment

The presence of critical limb ischaemia (CLI) was defined as either no palpable distal pedal pulses, TcPO2 < 30 mmHg [13,15] and/or arterial stenosis/occlusions documented by ultra-sound duplex or computed tomography or MRI requiring lower limb revascularization. The revascularization procedure was performed in respect to foot condition, ulcer location, vessels affected and patient’s general condition [13,16]. All subjects included in this study were treated by endovascular procedure (balloon angioplasty).

All patients were treated by dual antiplatelet therapy (acetylsalicylic acid 100 mg/day and clopidogrel 75 mg/day) which was started before the procedure and for at least one month after. In case of intolerance to acetylsalicylic acid or clopidogrel, ticlopidine (250 mg/day) was administered [16]. After discharge, patients were regularly followed (every 2–4 weeks) in our DFS until wound healing or different outcome.

Revascularization procedures were retrospectively analysed. Accordingly, in the case of successful foot revascularization defined by angiographic parameters (recanalization of pedal artery, or plantar arteries or both and presence of blood perfusion below-the-ankle documented by post-revascularization angiograms) or not, patients were respectively divided in two groups: successful foot perfusion (SFP) and failed foot perfusion (FFP).

### 2.5. Clinical Outcomes

Completed ulcer healing, minor and major amputation at 12 months of follow-up were evaluated. Definitive ulcer Healing was taken to be complete epithelialization of the target wound and maintenance of the closed healed epithelized surface for a minimum of 2 weeks. Minor amputation was defined any below-the-ankle amputation (digital, ray, metatarsal, Lisfranc, Chopart). Whilst major amputation was defined any amputation above the ankle. Outcomes for SFP and FFP were reported and compared. All potential predictors of outcomes were evaluated and described.

### 2.6. Statistical Analysis

Data were analyzed using the SPSS version 16.0 software (SPSS Inc, Chicago, IL, USA). The sample size was calculated by the power analysis by adopting the two-tailed test of the null hypothesis with α = 0.05 and a value of β = 0.10 as the second type error and, therefore, a test power equal to 90%. Continuous variables were expressed as the mean ± SEM. The Shapiro-Wilk test was used to statistically assess the normal distribution of the data. Comparisons between continuous variables were performed using the independent Student *t*-test or the Wilcoxon rank sum test. Categorical data were analyzed using the chi square test or the Fisher exact test. Univariable logistic regression analyses were performed for all potential predictor variables with the outcome of interest (healing, minor and major amputation), with values presented as univariable odds ratios (ORs) along with the respective 95% CI. All potential predictors were entered simultaneously into a multivariable logistic regression model. These models yielded a set of variables that best predict outcome. *p* < 0.05 was considered statistically significant

## 3. Results

Baseline demographic, clinical and vascular data are reported in Table 1. Eighty patients (80) were included. The mean age was 70.5 ± 10.9 years, 55 (68.7%) were male, 72 (90%) were affected by type 2 diabetes with a mean duration of 22.7 ± 11.3 years (Table 1). SFP showed less cases of heart failure (25.3 vs. 51.2%, *p* = 0.01) and dialysis (25.3 vs. 54.5%, *p* = 0.008) in comparison to FFP (Table 1).

### Outcomes

Overall 45 (56.2%) patients healed, 47 (58.7%) had minor amputation and 13 (16.2%) had major amputation at 1-year of follow-up. Outcomes for SFP and FFP were respectively: healing (89.3 vs. 9.1%, *p* < 0.0001), minor amputation (44.7 vs. 78.8%, *p* = 0.0001), major amputation (2.1 vs. 36.3%, *p* < 0.0001) (Figure 2).

At the multivariate analysis of predictors found at univariate analysis, failed foot revascularization resulted an independent predictor of non-healing, minor amputation, such as osteomyelitis and gangrene, and major amputation (Table 2).

## 4. Discussion

Even though it is recognized that BTA arterial disease increases the severity of PAD [11,12], there are few data about the effectiveness of foot revascularization in patients with ischaemic DFUs.

This study offers a specific overview on patients with ischaemic DFUs and BTA arterial and the impacts of foot revascularization. In the current study, the authors reported that patients who received successful foot revascularization had higher rate of healing (approximately 89 vs. 9%), lower rate of minor (approximately 45 vs. 79%) and major amputation (approximately 2 vs. 36%) in comparison to those with unsuccessful foot revascularization.

Overall, patients included in the study were aged and very fragile subjects with a long duration of diabetes (approximately 23 years), 76% had an ischaemic heart disease, 37% were under renal replacement therapy, 36% reported heart failure; in addition, they showed complicated DFUs: 85% had a gangrene, 81% were infected (60% had an osteomyelitis), and 72% an ulcer size > 5 cm^2^.

Nonetheless, the two groups reported similar characteristics at admission, even though FFP patients showed more cases of dialysis (54.5 vs. 25.3%) and heart failure (51.2 vs. 25.3%) compared to those with SFP.

Although heart failure and dialysis are recognized predictors of major amputation [17,18] in the current study these clinical conditions did not seem directly related to the outcomes of this specific group of patients. Otherwise, unsuccessful foot revascularization resulted an independent predictor of non-healing, minor and major amputation. In addition, the presence of osteomyelitis and gangrene was associated to minor amputation as already confirmed in other studies [19,20,21].

Successful foot revascularization allowed positive outcomes such as high rate of healing and very low rate of major amputation. Therefore, recanalization of foot arteries should be considered the main goal to achieve healing and limb salvage, and in patients with BTA arterial disease revascularization procedures should not be limited to infra-popliteal vessels.

However, our study showed also that a great number of patients with BTA arterial disease (approximately 40%) are not-treatable subjects reporting an unsuccessful foot revascularization procedure. FFP patients were in 50% on dialysis and it could justify the higher rate of revascularization failure as reported in previous studies [22,23].

In dialyzed persons with ischaemic DFUs, foot arteries, which provide direct blood flow to the wound, show very small diameters and, especially in the case of concomitant calcifications, are highly technically challenging to be treated and lead to high risk of failure [10].

The presence of heavily calcified vessel occlusions push the mechanical properties of the guidewire and balloon to their extreme limits and, the inability of the balloon catheter passage with the traditional antegrade techniques frequently leads to suboptimal procedural success rate of endovascular revascularization [10,11].

In the last years, the prevalence of PAD has been reported approximately in 65–70% of patients with ischaemic DFUs [24] being a current issue for professionals involved in the management of diabetic foot disease.

Among patients with ischaemic DFUs, approximately 25% are defined as no-option critical limb ischaemia (NO-CLI) which have a rate of 25–30% of major amputation due to the unsuccessful lower limb revascularization [22,23,25]. NO-CLI subjects are usually characterized by a multilevel arterial disease with the high involvement of foot arteries (approximately in 75% of cases) [23].

Nowadays, BTA arterial disease is a new frontier for clinicians, vascular surgeons and interventional radiologists/cardiologists managing diabetic foot disease. Foot revascularization is still under debate and a there is no common strategy. Although Huizing et al. have reported no significant difference in amputation free survival between BTA angioplasty and below-the-knee angioplasty [26], other studies have documented as BTA angioplasty is associated with reduced rate of major amputation [27,28,29] as we have found in our data.

Some studies have offered the feasibility and effectiveness of recanalization techniques for treating foot arteries in patients with BTA arterials: pedal-plantar loop, trans-collateral recanalization and retrograde percutaneous access have shown to be new useful options in patients not treatable by traditional antegrade access [30,31,32,33]. Plain old balloon angioplasty remains the standard technique in foot vessel angioplasty while stenting is contraindicated due to the high risk of mechanical trauma [29,34,35].

The main limit of these techniques is that sometimes an unsuccessful intervention may aggravate ischaemia, causing a deterioration of distal circulation which may be not suitable for further interventions.

Recently new techniques such as the deep venous arterialization (DVA) have been further explored especially for NO-CLI patients. As reported in preliminary studies, the DVA may be a future promising procedure to reduce amputation in NO-CLI subjects with BTA arterial disease [36,37,38,39]. The aim of DVA is to arterialize the target vein by an arteriovenous fistula increasing blood flow to the foot. This approach may promote the development of a new collateral network in the case of a “desert foot” and enhances neo-angiogenesis [40].

To our knowledge, the current study is the first one to evaluate specifically the effectiveness of foot revascularization in persons with ischaemic DFUs and BTA arterial disease. Even though a successful BTA revascularization reduces dramatically the rate of major amputation, there is nowadays a large part of NO-CLI patients in which foot revascularization is not technically feasible due to anatomical features and specific pattern of arterial disease. Nonetheless, future tools and procedures are promising to further improve current results.

This study is retrospectively analysed and data reported are based on the results of a single-centre where patients are managed by a specialized multi-disciplinary foot team, including expert interventional radiologists and vascular surgeons. The sample of patients included is not large. A larger sample could be useful to reinforce these data.

## 5. Conclusions

BTA arterial disease is the most severe pattern of PAD in persons with ischaemic DFUs. Despite the lack of published data, foot recanalization improves the outcomes and should be considered mandatory for managing patients with BTA arterial disease.

## Figures and Tables

**Figure 1 jcm-10-03977-f001:**
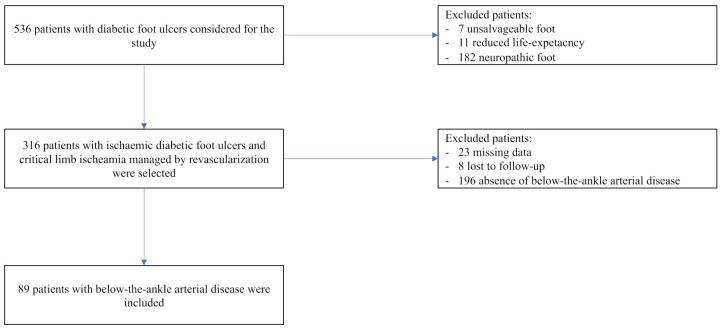
Flow-chart on patient’s recruitment and inclusion in the study.

**Figure 2 jcm-10-03977-f002:**
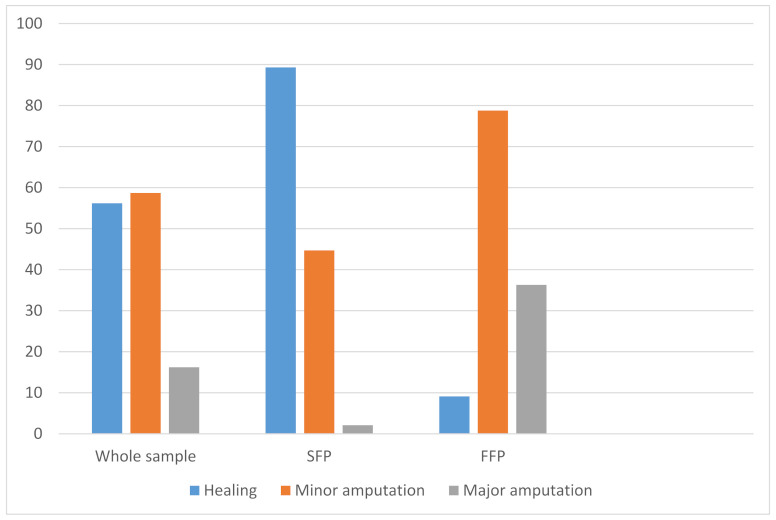
Outcomes for the whole sample, SFP, FFP. SFP: successful foot perfusion; FFP: failed foot perfusion.

**Table 1 jcm-10-03977-t001:** Baseline characteristic of whole sample, successful foot perfusion and failed foot perfusion groups. LDL: low-density lipoprotein; TAG: triglycerides; TcPO2: transcutaneous pressure of oxygen.

Variables	Whole Sample N = 89	SFP (*n* = 47)	FFP (*n* = 33)	*p*
Age (years)	70.5 ± 10.9	71.5 ± 10.3	69.3 ± 11.8	0.4
Sex (man)	55 (68.7%)	(30) 63.8%	(25) 75.7%	0.2
Diabetes (type 2)	72 (90%)	44 (93.6%)	28 (84.5%)	0.2
Diabetes duration (years)	22.7 ± 11.3	22.8 ± 10.7	22.6 ± 12.3	0.9
HbA1c % (mmol/mol)	7.8 ± 4.1 (62 ± 21)	7.9 ± 4.2 (63 ± 22)	7.8 ± 4 (62 ± 20)	0.1
Insulin therapy	65 (81.2)	39 (83%)	26 (78.8%)	0.5
Hypertension	68 (85%)	43 (91.5%)	25 (76.8%)	0.09
Dyslipidemia	62 (77.5%)	38 (80.8%)	24 (72.7%)	0.4
LDL (mg/dL)	90 ± 22	96 ± 25	81 ± 18	0.2
TAG (mg/dL)	165 ± 55	173 ± 65	154 ± 42	0.4
Smoke	11 (13.7%)	7 (14.9%)	4 (12.1%)	0.7
Ischaemic heart disease	61 (76.2%)	34 (72.3%)	27 (81.2%)	0.3
Heart failure	29 (36.2%)	12 (25.3%)	17 (51.2%)	0.01
Dialysis	30 (37.5%)	12 (25.3%)	18 (54.5%)	0.008
Cerebrovascular disease	35 (43.7%)	20 (42.5%)	15 (45.4%)	0.7
Ulcer size (>5 cm^2^)	58 (72.5%)	33 (70.2%)	25 (75.7%)	0.6
Previous lower limb revascularization	10 (12.5%)	6 (12.7%)	4 (12.1%)	0.9
Infection	65 (81.2%)	39 (82.9%)	26 (78.9%)	0.6
Gangrene	68 (85%)	38 (80.5%)	30 (90.1%)	0.2
Osteomyelitis	48 (60%)	28 (59.6%)	20 (60.1%)	0.9
C-Reactive Protein mg/dL)	66 ± 47	63 ± 50	72 ± 43	0.4
Baseline TcPO2 (mmHg)	16 ± 7	17 ± 8	14 ± 6	0.2
1-month post-procedure TcPO2 (mmHg)	40 ± 13	48 ± 9	29 ± 10	<0.0001

**Table 2 jcm-10-03977-t002:** Multivariate analysis of independent predictors of outcomes found at univariate analysis.

Variables	Non-Healing	Minor Amputation	Major Amputation
OR	95% CI	*p*-Value	OR	95% CI	*p*-Value	OR	95% CI	*p*-Value
Failed foot revascularization	8.5	1.8–15.8	0.0001	2.1	1.3–7.1	0.001	9.8	1.2–16.6	0.0001
Heart failure	0.8	0.4–1.2	0.3				1.1	0.7–1.3	0.09
Dialysis	0.9	0.5–2.1	0.1	0.6	0.4–1.1	0.8			
TcPO2 < 25 mmHg	1.3	0.7–1.5	0.09	1.5	0.9–2.3	0.06	2.3	1.2–5.7	0.03
Osteomyelitis				2.2	1.3–4.5.	0.02			
Gangrene				1.9	1.1–3.7	0.03			
Size (>5 cm^2^)				1.2	0.8–2.0	0.6

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
