# Peer review of "Foot Revascularization Avoids Major Amputation in Persons with Diabetes and Ischaemic Foot Ulcers"

_jcm, 2021, doi:10.3390/jcm10173977_

Round 1

Reviewer 1 Report

This is a retrospective study that assessed the impact of successful and failed revascularization procedures on wound healing and amputation rates. They included patients with diabetes and ischemic foot ulcers. The authors demonstrated that failed procedures were an independent predictor of amputations and healing of ulcers in people with ischemic DFU. I appreciate the authors for specifically reporting ischemic DFUs as this data is sparsely available in this field as most studies include/mix up neuropathic DFUs as well. Below are my comments to improve the paper.

Minor comments

  1. Ischemia is spelled differently throughout the manuscript. Please keep it the same as either ‘ischemia’ or ‘ischaemia’ as per the journal's English requirement.
  2. In the clinical assessment section, the authors use the term ‘considered’ which I think should be changed to ‘defined’. They are essentially defining the risk factors here. Otherwise it is misleading to be taken as an inclusion criteria.
  3. In table 1, can the authors provide the mean or median data for ulcer size of the included patients for each group?
  4. Can the authors report the number of previous revascularizations for the included patients in table 1?
  5. Can the authors also provide the medications taken by the patients? Eg., beta blockers, ARB, ACEI, antiplatelets, statins, aspirin, metformin.
  6. In table 1, ‘1-month TcPO2’ is not clear at the first read. I understand this is one month after the revascularization procedure. If so, could the authors change it to ‘1-month post procedure TcPO2’?

Major comments

  1. Could I recommend the authors to justify the study with strong case? Although the study is very important in this field, the introduction does not strongly support the reasoning for the study. Introduce about the specifics of why ischemic patients were specifically selected, prevalence of ischemia in these patient subset and how ischemic is a major contributing factor to amputation in these patients. Mention similar studies that were previously done and if amputation data was not reported there. As per reference 9 & 10, this topic was previously done. Emphasize on how this current study adds value to the existing information.
  2. The authors mention that the patients provided verbal consent. This is not in line with the declaration of Helsinki which advises the ethical principles for medical research involving human subjects. Can the authors explain why there was no documentation of consent? Although, I agree that institutional approval is sufficient for retrospective study. I would suggest removing the mention of verbal consent if it was not documented.
  3. Define SFP and FFP in clinical terms. How exactly it was decided based on the procedural parameters. Or was it based on clinician report. If so, mention this clearly and discuss it as a limitation.
  4. In the statistical analysis section, authors mention that data were expressed as mean and SD. I believe this was for continuous data only. Could the authors expand further on this – Were normality of data checked? What tests were used? How was non-normally distributed data handled? If non-normally distributed, could the authors report them as median and IQR? How were categorical data presented? How were nominal and non-parametric data tested?
  5. In the logistic regression analysis, the authors report that univariate analyses of all predictor variables were done first to determine the independent predictors. During the multivariate analyses, how did the authors handle collinearity? Could the authors explain the steps followed to avoid overfitting of the model? Did they calculate the deviance for each variable in the model? Was there any interaction present between any variables?
  6. If the inclusion criteria is ischemic DFU, how come only 90% patients had diabetes? Shouldn’t it be 100%? Please explain.
  7. Can the authors provide the type of revascularization procedures performed in the included patients? eg., stent, balloon angioplasty, drug coated, etc. Also, mention the success and failed procedures for the specific revascularization procedures.
  8. Can the authors provide the LDL, TAG levels? Reason being - were patients with failed procedures be non responders? Could there be an association between failed procedures and non responders? 
  9. Can the authors provide the Rutherford category of the included patients? Was category 5 associated with higher number of major amputation? If so, include this variable in the regression model.
  10. Table 1 shows hypertension to be significantly different between the groups. However, the authors have not included this variable in the regression model. Could the authors explain why? Also, could the authors explain the basis for covariate selection in the methods section?
  11. A recent NMA showed that revascularization procedure in claudication patients is beneficial for short term only, and no beneficial effects can be seen in long term. How do the authors relate this to their ischemic patients? https://www.ahajournals.org/doi/full/10.1161/JAHA.120.019672

Author Response

The Authors all say many thank for this revision. Comments and suggestions have improved the text and better focused on the aim of this study. 

Minor comments

  1. Ischaemia has been now used in all cases;

  1. “Defined” instead of “considered” has been now used as suggested;

  1. The Authors do not have this data. In this retrospective study we have only the data on ulcer size (< or > 5 cm2) reported at the first assessment. This cut-off has been used because it has been document that ulcers >5 cm2 have reduced chance of healing in comparison to smaller ulcers

  1. The data has been reported as suggested (see Table 1);

  1. Unfortunately, the Authors do not have all the medications taken by the patients. Anyway, as explained in the manuscript, all patients with hypertension and dyslipidemia were treated by anti-hypertensive therapy and statin respectively, and all patients were treated by antiplatelet therapy (dual antiplatelet therapy for at least one month after revascularization). According to available data, we have now added who were treated by insulin therapy or not (see table 1);

  1. To be clearer “1-month post-procedure TcPo2” has been used as suggested;

Major comments

  1. The introduction has been revised to better justify and support the reasoning fort this study as suggested (see lines 57-68);

  1. The text has been modified as suggested;

  1. SFP and FFP has been better defined. As reported now in the text, successful foot perfusion and failed foot perfusion were defined by angiographic/vascular parameters (successful or unsuccessful revascularization of BTA arteries and presence or not of BTA blood perfusion after revascularization documented by post-procedure angiogram). See lines 126-128

  1. The statistical analysis section has been revised and better described.

  1. The statistical analysis section has been revised and better described.

  1. All patients included had diabetes. In the table 1, “90%” were referred to patients with type 2 diabetes

  1. All revascularization of arteries below-the-ankle were performed by balloon angioplasty. The data has been added in text. See line 119.

  1. Low-density-lipoprotein (LDL) and triglycerides (TAG) values reported at the first assessment have been now included in the table 1. Anyway, there is not association between LDL or TAG values and failed procedures.

  1. All patients included belonged to category 5 of Rutherford classification.

  1. Hypertension has been included in the variable regression model, anyway it did not show any significant impact on outcomes of interest.

  1. All patients included in this study received revascularization procedure due to presence of non-healing ulcer, minor tissue loss, gangrene and concomitant critical limb ischaemia. There were very few case of patients affected by symptomatic claudication and it has not been the reason for performing revascularization procedure.

Reviewer 2 Report

I Liked it a lot. I treat thousands of diabetics with ulcer and PVD is a significant cause of non healing and amputation. it is very difficult to remedy. we need more of these studies to show the mobility and mortality of this disease that affects billions of people 

Author Response

No revision required.

Round 2

Reviewer 1 Report

I thank the authors for addressing all the comments. I would be happy to accept the manuscript in this updated version.